# A Comparative Analysis of Chemical Constituents and Antioxidant Effects of *Dendrobium fimbriatum* Hook Fractions with Different Polarities

**DOI:** 10.3390/ijms241612646

**Published:** 2023-08-10

**Authors:** Tianrui Zhao, Fangyuan Zheng, Yaping Liu, Afsar Khan, Zhengxuan Wang, Guiguang Cheng

**Affiliations:** 1Faculty of Food Science and Engineering, Kunming University of Science and Technology (KUST), Kunming 650500, China; food363@163.com (T.Z.); zfy199836@163.com (F.Z.); liuyaping@kust.edu.cn (Y.L.); 2Department of Chemistry, COMSATS University Islamabad, Abbottabad Campus, Abbottabad 22060, Pakistan; afsarkhan@cuiatd.edu.pk

**Keywords:** *Dendrobium fimbriatum* Hook, polar fractions, antioxidant, chemical constituents, Nrf2

## Abstract

The aim of this study was to investigate the chemical composition and antioxidant capacity of various polar fractions obtained from *Dendrobium fimbriatum* Hook (DH). First, a 90% ethanol-aqueous extract of DH (CF) was subjected to sequential fractionation using different organic solvents, resulting in the isolation of a methylene chloride fraction (DF), an ethyl acetate fraction (EF), an n-butanol fraction (BF), and a remaining water fraction (WF) after condensation. Additionally, the CF was also subjected to column chromatography via a D101 macroreticular resin column, eluted with ethanol-aqueous solution to yield six fractions (0%, 20%, 40%, 60%, 80%, and 100%). UPLC-Q-Exactive Orbitrap-MS/MS analysis identified a total of 47 chemical compounds from these polar fractions, including fatty acids, amino acids, phenolic acids, flavonoids, organic heterocyclic molecules, and aromatic compounds. Moreover, DF, EF, and the 60%, 80%, and 100% ethanol-aqueous fractions had higher total phenol content (TPC) and total flavonoid content (TFC) values and greater 2,2′-azinobis (3-ethylbenzothiazoline-6-sulfonic acid) diammonium salt (ABTS-) and 1,1-diphenyl-2-picrylhydrazyl (DPPH)-scavenging abilities. In H_2_O_2_-induced HepG2 cells, the aforementioned fractions could increase the activities of antioxidative enzymes NAD(P)H: quinone oxidoreductase 1 (NQO1), superoxide dismutase (SOD), heme oxygenase-1 (HO-1) and catalase (CAT), stimulate glutathione (GSH) synthesis by increasing the activities of glutamic acid cysteine ligase (GCL) and glutathione synthetase (GS), regulate GSH metabolism by increasing glutathione peroxidase (GSH-Px) and glutathione reductase (GR) activities, and reduce levels of reactive oxygen species (ROS) and malondialdehyde (MDA). Furthermore, the antioxidative stress effect of the DH fractions was found to be positively correlated with the activation of nuclear factor-erythroid 2-related factor 2 (Nrf2) protein and the presence of antioxidative chemical constituents. In conclusion, this study highlights the efficacy of both liquid–liquid extraction and macroporous resin purification techniques in the enrichment of bioactive compounds from natural food resources. The comprehensive analysis of chemical constituents and antioxidant effects of different polar fractions from *Dendrobium fimbriatum* Hook contributes to the understanding of its potential application in functional foods and nutraceuticals.

## 1. Introduction

*Dendrobium fimbriatum* Hook (DH) (family Orchidaceae) is an edible and ornamental plant, which is mainly distributed in the Yunnan and Guangxi Provinces of China [1,2]. As a unique, healthy edible plant, DH is generally used as an herbal tea, soup ingredient, or a juice for the protection of the liver, lung, spleen, stomach, etc. [3,4]. It has been reported that DH is rich in polysaccharides, flavonoids, polyphenols, fatty acids, amino acids, and so on, showing good antioxidative, anti-inflammatory, hypotensive, and hepatoprotective characteristics [4,5,6]. In recent years, some studies have revealed that DH extract has important regulatory effects on cancers, metabolic disorders, and immunological diseases, but the compounds responsible for these effects were not clear [3,7]. Therefore, it is meaningful to investigate the bioactive agents of DH extract for their health applications.

Oxidative stress induced by reactive oxygen species (ROS) plays a regulatory role in the initiation, development, and evolution of many diseases, such as neurodegenerative disease, cardiovascular disease, chronic liver disease, type 2 diabetes, aging, and so on [8,9]. Currently, natural phenolic acids and flavonoids, such as curcumin, 6′-*O*-caffeoylarbutin, and dihydrochalcone, are important natural antioxidants, which could effectively improve the activities of endogenous antioxidant enzymes to suppress ROS-induced oxidative stress [10,11,12]. Furthermore, some studies have focused on the enrichment or purification of natural antioxidants from teas, flowers, fruits, and even medicinal and edible plants [13,14]. As reported elsewhere, physical and chemical properties, including polarity, solubility, ionization, and so on, are vital factors in enrichment or purification techniques [15,16]. Liquid–liquid extraction and macroporous resin purification techniques are commonly used to enrich or purify bioactive constituents [16]. For example, the liquid–liquid extraction method has been used for the preparation of different polar fractions to obtain an antioxidant extract with a high phenolic content from artichokes [17]. The macroporous resin purification technique is another efficient method for the enrichment of antioxidants from plant resources for its high adsorption capacity [15].

To date, there has been no study of the enrichment or purification of bioactive agents from *Dendrobium fimbriatum*. Therefore, this paper aims to compare the enrichment or purification effects of liquid–liquid extraction and macroporous resin purification techniques on antioxidant compounds from DH. The evaluation of their antioxidant and regulatory effects on H_2_O_2_-induced HepG2 cells is also performed in this study.

## 2. Results

### 2.1. Identification of Chemical Constituents in Different DH Fractions

The bioactivities of natural plants are dependent on their chemical compositions [18,19,20]. More than 2300 compounds have been identified (Appendix A) in DH extract, and these chemical compounds could be classified into over eighteen different types according to their structures (Figure 1A). Among these, fatty acids (206 compounds) accounted for the largest proportion, followed by organic heterocyclic compounds (181 compounds), amino acids (178 compounds), and alkaloids (100 compounds). It is worth noting that flavonoids and isoflavones (129 compounds) accounted for 5.6% of all compounds. After these compounds were matched against a reliable mass spectroscopic database, the detailed information regarding the 47 main chemicals of the DH fractions was summarized and compiled in Table 1, including relative content, retention time, *m*/*z* value, and molecular formula. The UpSet graph seen in Figure 1B displays the 11 constituents found in all DH polar fractions and the 5 chemical constituents found in in ten DH fractions. The results reveal that the enrichment technique was an important factor in determining the chemical compositions of DH fractions.

### 2.2. The Values of TPC and TFC in DH Fractions

Phenolics and flavones are specific substances among natural products, which show bioactivity in many studies [17,19,21]. Thereby, the levels of total phenolic content (TPC) and total flavone content (TFC) were measured in this study. The results (Table 2) showed that the values of TPC and TFC in eleven DH fractions were from 23.32 ± 2.39 to 298.13 ± 1.59 mg gallic acid equivalent (GAE)/g extract and from 6.05 ± 1.04 to 47.05 ± 1.73 mg RE/g extract, respectively. Particularly, the TPC and TFC contents in EF showed the highest contents of 298.13 ± 1.59 mg GAE/g extract and 47.05 ± 1.73 mg rutin equivalent (RE)/g extract, respectively. Differences were also found in 0–100% ethanol-aqueous fractions. Their results showed that TPC and TFC contents were increased with concentrations of ethanol-aqueous solution (0–80%), showing significantly positive relations (r = −0.935, *p* < 0.05). These findings suggested that different enrichment techniques could obviously affect the contents of TPC and TFC in DH fractions.

### 2.3. ABTS and DPPH Radical Scavenging Abilities of DH Fractions

The scavenging abilities of the 2,2′-azinobis (3-ethylbenzothiazoline-6-sulfonic acid) diammonium salt (ABTS) and 1,1-diphenyl-2-picrylhydrazyl (DPPH) radicals are generally used for evaluating the antioxidative capacity of natural products. The results (Table 3) showed that various DH fractions had scavenging abilities on ABTS and DPPH radicals, especially ethyl acetate fraction (EF) and dichloromethane fraction (DF),the 60–100% ethanol-aqueous fractions had lower IC_50_ values of scavenging ABTS and DPPH radicals. This may be related to their higher levels of TPC and TFC. According to the relationships of TPC and TFC with the antiradical assay and chemical compositions in Figure 2, the DH fractions of DF; EF; and 60%, 80%, and 100% ethanol-aqueous fractions had better scavenging abilities on free radicals due to their higher levels of TPC and TFC. Therefore, these five fractions were selected for further cellular experimental studies.

### 2.4. Effect of DH Fractions on the Cell Viability and Oxidative Stress in HepG2 Cells

To explore the cytotoxicity of DH fractions on HepG2 cells, the cell viability was performed by MTT assay. As shown in Figure 3A, the DH fractions (DF, EF, and 60%, 80%, and 100% ethanol-aqueous fractions) with the 100 μg/mL concentration showed more than 90% viability of HegG2 cells, which was higher than that of the 200 μg/mL concentration. Thereby, the concentration of 100 μg/mL was selected for further studies.

ROS are critical inducers for MDA production in cells, while both ROS and MDA cause oxidative stress [17]. Therefore, the contents of ROS and MDA were determined in this study. In comparison with the control group, there was a significant increase in ROS level in H_2_O_2_ group, showing an abundant accumulation of ROS (Figure 3B). With pretreatment by DF, EF, and 60%, 80%, 100% ethanol-aqueous fractions, the ROS content was dramatically reduced compared with the H_2_O_2_ group by 33.60%, 34.64%, 30.63%, 38.33%, and 43.18%, respectively (*p* < 0.05). It is worth mentioning that the ROS content of the 100% ethanol fraction was lower than that of the VC group. Similarly, the MDA content (Figure 3C) in H_2_O_2_ group was markedly higher than in the control group by 248.57% (*p* < 0.05). It was scavenged by pretreatment with DF, EF, and 60%, 80%, 100% ethanol-aqueous fractions by degrees of 22.45%, 30.82%, 40.37%, 33.17%, and 25.63%, respectively (*p* < 0.05). The results showed that five DH fractions had an antioxidative capacity by reducing the accumulation of ROS and MDA in H_2_O_2_-induced HepG2 cells. 

### 2.5. Influence of DH Fractions on GSH Synthesis and Metabolism

GSH is the most abundant antioxidant, and it is important throughout the body [22,23,24]. It is an effective strategy to increase antioxidative ability by promoting the synthesis and metabolism of GSH. Hence, the activities of glutamic acid cysteine ligase (GCL), glutathione synthetase (GS), glutathione peroxidase (GSH-Px), glutathione reductase (GR), and the GSH level were determined in this study. As shown in Figure 3, the H_2_O_2_ treatment sharply decreased the GSH content and the activities of GCL, GS, GR, and GSH-Px by 31.85%, 47.96%, 57.23%, 63.22%, and 35.99%, respectively. Fortunately, upon pretreatment with DF, EF, and 60%, 80%, 100% ethanol fractions, the GSH contents (Figure 4A) were significantly increased by 26.89%, 40.05%, 26.10%, 34.24%, and 43.85% (*p* < 0.05), respectively. Similarly, the activities of antioxidative enzymes GCL (Figure 4B), GS (Figure 4C), GR (Figure 4D), and GSH-Px (Figure 4E) were markedly enhanced by pretreatment with these DH fractions. In particular, EF exerted a better antioxidative effect than DF on activating the oxidation–reduction cycle of GSH, while the 100% ethanol fraction exhibited the highest effect on GSH synthesis and metabolism, which was close to the VC group.

### 2.6. Antioxidative Activities of SOD, CAT, HO-1, and NQO1

The antioxidative enzyme system, consisting of SOD, CAT, HO-1, NQO1, and so on, is responsible for the protection of organisms against oxidative stress [22,25]. Hence, improving the activities of antioxidative enzymes is an effective strategy for decreasing the accumulations of ROS and MDA. The present results (Figure 5) revealed that DH fractions (DF, EF, and 60%, 80%, 100% ethanol-aqueous fractions) effectively reversed the decreasing activities of SOD (Figure 5A), CAT (Figure 5B), HO-1 (Figure 5C), and NQO1 (Figure 5D) induced by H_2_O_2_ in HepG2 cells (*p* < 0.05). In particular, the EF and 80% and 100% ethanol-aqueous fractions enhanced the activities of these antioxidative enzymes. This finding supported that these five DH fractions could reduce ROS and MDA accumulations by increasing the activities of SOD, CAT, HO-1, and NQO1.

### 2.7. Activation of Nrf2 Expression in H_2_O_2_-Induced HepG2 Cells by DH Fractions

Nrf2 is a very important antioxidative regulator that can transfer from cytoplasm to the cell nucleus to bind with the antioxidant response element (ARE), which then initiates the transcription of antioxidative enzymes [25]. Thereby, the total Nrf2 protein expression was determined by Western blotting, and the nuclear protein expression of Nrf2 was observed by fluorescence staining. The result (Figure 6A) showed that H_2_O_2_ markedly reduced total Nrf2 expression in comparison with the control group by 38.93% (*p* < 0.05). A significant finding was that the pretreatments with DF, EF, and 60%, 80% and 100% ethanol-aqueous fractions caused a dramatic increase in total Nrf2 protein expressions by 40.03%, 69.52%, 54.49%, 62.38%, and 77.98%, respectively. The 4′,6-diamidino-2-phenylindole (DAPI) staining result (Figure 6B) showed that the nuclear Nrf2 expression was also increased by DH fractions, especially in EF and the 80% and 100% ethanol-aqueous fraction groups. These results demonstrated that DH fractions could increase the total Nrf2 protein levels and expressions of nuclear Nrf2 protein. 

### 2.8. Multivariate Analysis

The principal component analysis (PCA) extracted from the data of Table 2 and Table 3 and Figure 3, Figure 4, Figure 5 and Figure 6 exhibited 88.4% of total variation, where principal component 1 (PC1) accounted for 70.9% of the variance and principal component 2 (PC2) accounted for 17.5% (Figure 7A). Moreover, there were positive relationships of the DH fractions (EF and 80% and 100% ethanol fractions) and the activities of antioxidative enzymes NAD(P)H: quinone oxidoreductase 1 (NQO1), superoxide dismutase (SOD), heme oxygenase-1 (HO-1), and catalase (CAT), GSH synthetic and metabolic enzymes, and Nrf2 protein, suggesting that these three DH fractions exerted better antioxidative ability. The Venn diagram result (Figure 7B) showed that there were 30 constituents among five DH fractions, and the 80% ethanol extraction had the highest contents of 44 constituents, followed by 42 constituents in EF and the 60% and 100% ethanol fractions. The DF had only 42 constituents, which may be associated with its weaker antioxidative capacity compared with other groups. The relationship net of chemical components and bioactivities in Figure 7C demonstrated that the antioxidative capacity was the most common bioactivity of these constituents, followed by anti-inflammation, antitumor effect, antibacterial activity, etc. Therefore, the excellent antioxidative ability of EF and the 80% and 100% ethanol fractions of DH were close to their higher content of antioxidative constituents.

## 3. Discussion

It has been demonstrated that the bioactivities of natural extracts are dependent on their chemical composition [26,27,28]. To date, studies on DH have been focused mainly on its polysaccharide, but the analysis of its chemical composition with small molecules has not received attention. Only a few studies reported that DH has flavonoids that show a high activity on scavenging free radicals in vitro [6,7]. Thereby, with the aim to systematically study and comprehensively compare the chemical compositions and antioxidant abilities of DH, liquid–liquid extraction methods and purification techniques were performed in this study. The results revealed that liquid–liquid extraction and macroporous resins purification techniques significantly affected the chemical constituents and antioxidant abilities of DH fractions.

Flavonoids and polyphenols are found in many plants, flowers, or herbs and have a very important regulatory effect on bioactivities such as antioxidant ability, anti-inflammation, anti-apoptosis, and so on [17,29]. Some studies have reported that extraction methods or purification techniques could affect chemical composition and proportion [15,17]. A similar result was also found in this work, where the TPC and TFC levels in 11 DH fractions using liquid–liquid extraction and macroporous resin purification techniques showed significant differences. The DF, EF, and 60%, 80%, and 100% ethanol-aqueous fractions had higher levels of TPC and TFC, indicating that the chemical constituents of DH extracts could be enriched or fractioned by various solubilities and polarities. Of course, these five fractions with high TPC and TFC levels exhibited better ABTS and DPPH radical scavenging abilities. Additionally, the 80% ethanol-aqueous fraction showed the best scavenging effect on ABTS radicals, and EF exhibited the best capacity on DPPH radical scavenging. This may be related to differences in lipid-soluble and water-soluble chemical constituents of EF and the 80% ethanol-aqueous fraction because the ABTS radical is a water-soluble radical, and DPPH radical has better lipid solubility. Therefore, various solubilities or polarities were important factors of DH fractions on scavenging different radicals.

The antioxidant system, consisting of antioxidants and antioxidative enzymes, exerts a vital regulatory role in maintaining balance between oxidation and reduction [30,31,32]. As reported, the SOD, CAT, and GSH-Px enzymes are the first defensive line to scavenge ROS, such as the free radicals of $O_{2}^{\cdot -}$, H_2_O_2_, and HO^·^ [33,34]. With the catalyzation of SOD, the $O_{2}^{\cdot -}$ can be transformed into H_2_O_2_, and the latter will be resolved into water and O_2_ under the catalyzation of CAT [33,34]. Additionally, hydroperoxide (MDA) or HO^·^ will be converted into an innocuous substance or H_2_O_2_ by GSH-Px [35,36]. Therefore, these three antioxidative enzymes exerted an antioxidative effect by scavenging ROS synergistically. Moreover, HO-1 and NQO1 can effectively alleviate ROS-induced oxidative stress damage [34]. Antioxidant ability is a vital character for many natural extracts, for example, E Se tea extracts, Que Zui tea extracts, *Anneslea fragrans* extract, etc., which exhibited obvious antioxidant ability by enhancing antioxidative enzymes (SOD, CAT, NQO1, HO-1, etc.) [17,29,30]. The same results were also obtained in this study, which showed that DH fractions (DF, EF, and 60–100% ethanol-aqueous fractions) could effectively enhance the activities of SOD, CAT, HO-1, NQO1, and GSH-Px to reduce the accumulation of ROS and MDA.

GSH is the most important and abundant antioxidant to construct an endogenous non-enzymatic antioxidant system in organisms [23,37]. Its synthesis is mainly limited to the synthetic enzymes GCL and GS, among which the former is the first rate-limiting enzyme for GSH synthesis [38]. It has been verified that peroxidative products such as MDA are commonly degraded by GSH, which is assisted by the catalyzation of GSH-Px [39]. On the other hand, the oxidized GSH (GSSG) is reduced to GSH via the catalyzation of GR, indicating that GSH exerts an antioxidative effect via its own redox reaction with the catalyzation of GR and GSH-Px [37]. Therefore, promoting the synthesis and metabolism of GSH is a very important safeguard for reducing ROS- or MDA-induced oxidative stress. In this study, we found that DH fractions (DF, EF, and 60%, 80%, and 100% ethanol fractions) promoted the redox reaction of GSH by increasing GCL, GS, GR, and GSH-Px activities, which supported the above results of reducing ROS and MDA accumulations.

To demonstrate the antioxidative mechanism of DH fractions on increasing endogenous antioxidant capacity, the nuclear factor erythroid 2-related factor 2 (Nrf2) pathway should not be ignored. As a key factor for the suppression of oxidative stress, the Nrf2 pathway has a central regulatory effect on maintaining the balance between oxidation and reduction in the body [40,41,42]. Nrf2 is recognized as a sensor of oxidative stress, which rigorously regulates the endogenous antioxidative defense system in combination with the original antioxidant response element (ARE) to initiate the downstream expressions of antioxidants (GSH) and antioxidative enzymes (SOD, CAT, HO-1, and NQO1) [43]. Additionally, it has been demonstrated that activating the Nrf2 pathway could effectively promote the synthesis and metabolism of GSH by increasing the activities of GCL, GS, GR, and GSH-Px [42]. The antioxidant and antioxidative enzymes regulated by Nrf2 are responsible for inhibiting the formation of and scavenging the accumulation of ROS and MDA. Our results (Figure 6) showed that DH fractions, especially EF and 80% and 100% ethanol-aqueous fractions, increased the Nrf2 protein expression and promoted the immigration of the Nrf2 protein into cell nucleus. These results indicated the potential antioxidative mechanism of DH fractions, which may be related to the activation of Nrf2 pathway. This may be associated with their chemical constituents (Figure 7A), which showed that there were 30 common chemical constituents (Figure 7B) in these five different DH fractions. The network of chemical compositions and bioactivities (Figure 7C) further revealed that 83.33% of these total 30 chemical constituents, including gastrodin, isovitexin, rosmarinic acid, etc., exhibited an antioxidant effect. These chemicals were reported to increase antioxidative capacity in vitro and in vivo by activating the Nrf2 pathway to enhance the activities of antioxidative enzymes (SOD, CAT, HO-1, NQO1, etc.) and promoting GSH synthesis [43,44,45]. These chemical constituents with antioxidants may be the foundation for DH to reduce oxidative stress by activating the Nrf2 pathway and to increase the endogenous antioxidant capacity.

Conclusively, various antioxidative abilities of different polar fractions of DH by liquid–liquid extraction and macroporous resin purification techniques are associated with their own chemical compositions and proportions. In this study, different DH fractions and antioxidant abilities were for the first time concatenated to select a better extraction or enrichment technique for enhancing the antioxidant ability of DH extract, which may support an insight that the bioactive effect of crude extracts from plants, flowers, or herbs could be strengthened by various liquid–liquid extraction and macroporous resin purification techniques. Moreover, an interesting phenomenon was found that, except for antioxidative ability, various DH fractions showed other bioactivities such as anti-inflammation, anti-depression, immunoregulation, etc., suggesting that different DH bioactivities may need various enrichment or purification techniques rather than only one. Of course, more studies are required to verify this hypothesis.

## 4. Materials and Methods

### 4.1. Chemicals and Reagents

2,2′-azinobis(3-ethylbenzothiazoline-6-sulfonic acid) diammonium salt (ABTS), 1,1-diphenyl-2-picrylhydrazyl (DPPH), 3-(4,5-dimethyl-2-thiazolyl)-2,5-diphenyl-2-H-tetrazolium bromide (MTT) reagents, 2′,7′-dichlorofluorescin diacetate (DCFH-DA), foline-phenol, and trypsin were purchased from Sigma-Aldrich (Shanghai, China). Human hepatocellular carcinoma cell line (HepG2) was purchased from Kunming Cell Bank (Chinese Academy of Sciences, Kunming, China). Fetal bovine serum (FBS), penicillin, streptomycin, and Dulbecco’s modified Eagle’s medium (DMEM) were purchased from Beijing Solaibao Technology Co., Ltd. (Beijing, China). The kits of superoxide dismutase (SOD), malondialdehyde (MDA), glutathione (GSH), glutathione peroxidase (GSH-Px), glutathione S-transferase (GST) were purchased from Nanjing Jiancheng Bioengineering Research Institute (Nanjing, China).

### 4.2. Preparation of Dendrobium fimbriatum Hook Fractions

The stems of *Dendrobium fimbriatum* were collected from Baoshan city in Yunnan Province of China. The samples were freeze-dried by a lyophilizer, crushed, and then extracted by 90% ethanol-aqueous solution using ultrasonic assistance extraction method three times. After centrifugation, the supernatant was collected for concentration using a rotary evaporator (Heidolph, Schwabach, Germany) to obtain the crude extract (CF). The CF was further fractionated by two methods as follows (Figure 8):(a)The CF powder (2 g) was fractionated with dichloromethane, ethyl acetate and n-butanol (each 3 times) in sequence. After rotary evaporation and lyophilization, the dichloromethane (DF, 336 mg), ethyl acetate fraction (EF, 100 mg), n-butanol fraction (BF, 691 mg), and the remaining water fraction (WF, 784 mg) were obtained for further studies.(b)The CF powder (2 g) was subjected to a column filled with D101 macroporous adsorption resin, which was eluted by different polarities of ethanol-aqueous solutions, to obtain 0% fraction (672 mg), 20% fraction (215 mg), 40% fraction, (216 mg), 60% fraction (174 mg), 80% fraction (152 mg), and 100% fraction (126 mg).

### 4.3. Chemical Constituent Analysis of DH Fractions

Chemical profiling of different DH extracts was analyzed using ultra-high performance liquid chromatography tandem hybrid quadrupole-orbitrap mass spectrometry (UPLC-Q-Exactive Orbitrap-MS/MS). Briefly, different polar fractions of DH were dissolved in ethanol solution to prepare standard solution and were filtered by 0.22 μm organic membrane for further analysis. The samples (2.0 μL) were analyzed by an UPLC-Q-Exactive Orbitrap-MS/MS equipment with a reverse-phase C18 column (Waters Technologies, Shanghai, China, 2.1 × 100 mm, 1.8 μm) at 0.2 mL/min flow rate and 35 °C column temperature. The mobile phases consisted of solvent A (ultra-pure water containing 5 mmol/L ammonium acetate and 5 mmol/L acetic acid) and solvent B (acetonitrile). The chromatographic conditions were 0–0.7 min for 1% B; 0.7–11.8 min for 99% B, 11.8–12 min for 99–1% B; 12–15 min for 1% B. Mass spectra were obtained by heating an electrospray ionization source in negative ion mode.

The key parameters of mass spectrometry were as follows: spray voltage was +3.8 and −3.4 kV; sheath gas flow rate was 50 arbitrary units (arb); auxiliary gas flow rate was 15 units (arb); capillary temperature was 320 °C; and auxiliary gas heater was 350 °C. The scanning mode was full scan with a 60,000 resolution, and the resolution of data-related MS/MS was 15,000. The charge transfer and normalized collision energies were 10, 30, and 60 eV. The data acquisition and processing were carried out using the software Xcalibur 4.1 (Thermo Scientific, Waltham, MA, USA).

### 4.4. Determinations of TPC and TFC

The methods used for the determinations of total polyphenol content (TPC) and total flavonoid content (TFC) are already described in our previous study [18]. Briefly, the sample solutions (DF, EF, BF, WF, and 0%, 20%, 40%, 60%, 80%, and 100% fractions) (0.2 mg/mL, 0.2 mL) were dissolved in 80% ethanol-aqueous solution, which was then mixed with folin phenol regent (0.1 mL) for 1 min, and then incubated with Na_2_CO_3_ (20% m/v, 0.3 mL) and 1.2 mL ultra-pure water at 70 °C in a water bath for 10 min. The absorbance of the mixture was then determined at 765 nm. The TPC content was expressed in gallic acid equivalents (mg GAE/g extract).

As for the measurement of TFC content, the sample (0.2 mg/mL, 0.3 mL), ethanol solution (60%, 0.95 mL), and NaNO_2_ solution (10% m/v, 75 μL) were mixed for 8 min. Then, to the mixture was added 75 μL Al(NO_3_)_3_ solution and 1.0 mL ethanol solution (60%). The solution was incubated for 12 min, and the absorbance was measured at 510 nm. The TFC content was expressed in rutin equivalents (mg RE/g extract).

### 4.5. Determination of Antioxidant Activities of DH Fractions

#### 4.5.1. ABTS Scavenging Assay

ABTS free radicals were produced by mixing ABTS stock solution (7 mM) with potassium persulfate (2.45 mM), and the mixture was incubated for 12–16 h in the dark. The ABTS solution was diluted with 0.2 M phosphate buffer (pH 7.4) to an absorbance level of 0.70 ± 0.02 at 734 nm. The ABTS scavenging activity was determined by mixing 10 μL of the 11 DH fractions of DF, EF, BF, WF, and 0%, 20%, 40%, 60%, 80%, and 100% fractions (62.5, 125, 250, 500, and 1000 μg/mL) with 990 μL ABTS solution. After a 10 min reaction, the absorbance was measured at 734 nm. The scavenging capacity of ABTS was calculated as follows:$$\mathrm{Scavenging}\;\mathrm{rate}\;\mathrm{of}\;\mathrm{ABTS}\;\left(\%\right)=\left[\frac{A_{\mathrm{control}}-A_{\mathrm{sample}}}{A_{\mathrm{control}}}\right]\times 100\%$$

#### 4.5.2. DPPH Scavenging Assay

The samples of DF, EF, BF, WF, and 0%, 20%, 40%, 60%, 80%, and 100% fractions (62.5, 125, 250, 500, 1000 μg/mL, and 1 mL) and DPPH solution dissolved in 95% ethanol (0.1 mM and 1 mL) were mixed by shaking and then incubated at room temperature in the dark for 30 min. The absorbance was determined at 517 nm. The scavenging capacity of DPPH was calculated as follows:$$\mathrm{Scavenging}\;\mathrm{rate}\;\mathrm{of}\;\mathrm{DPPH}\;\left(\%\right)=\left[\frac{A_{\mathrm{control}}-A_{\mathrm{sample}}}{A_{\mathrm{control}}}\right]\times 100\%$$

### 4.6. Estimation of DH Fractions on HepG2 Cell Toxicity

The HepG2 cells (1 × 10^5^ cells/mL, 200 μL) were seeded into a 96-well plate for 24 h incubation. The MTT assay was used to detect the toxicity of DH fractions (DF, EF, and 60%, 80%, and 100% fractions) with different polarities. The control group was treated with 200 μL DMEM, and experimental groups were treated with 200 μL DH fractions of different concentrations (50, 100, and 200 μg/mL). After incubation for 24 h, MTT solution (0.5 mg/mL, 150 μL) was added to all wells, incubated for 4 h, and then mixed with DMSO for 5 min. The absorption was determined at 570 nm using a microplate reader (SpectraMax M5, Molecular Devices, San Jose, CA, USA).

### 4.7. Determination of ROS Content in H_2_O_2_-Induced HepG2 Cells

The measurement of ROS in H_2_O_2_-induced HepG2 cells was carried out according to the method described by Zhao with minor modifications [18]]. HepG2 cells were grown for 12 h with a density of 1 × 10^5^ cells/mL in a 6-well plate. The sample groups (100 μg/mL of 5 DH fractions, 1 mL), the VC group (100 μg/mL VC, 1 mL), and the H_2_O_2_ group (1 mL DMEM) were mixed with 1.0 mmol/L H_2_O_2_ for 24 h, while the control group was treated only with 2 mL DMEM solution for 24 h. After digestion by trypsin, 1 mL of PBS was added to the cells for the collection of cell supernatants. The cell supernatants were then incubated with 1 mL of DCFH-DA (10 μM) at 37 °C for 30 min in the dark. The intracellular green fluorescence intensity was measured at 485 nm excitation and 525 nm emission wavelengths by flow cytometry (Guava^®^ easyCyte 6-2L, Millipore, Billerica, MA, USA).

### 4.8. Measurements of GSH Synthesis and Metabolism in H_2_O_2_-Induced HepG2 Cells

The methods for cell culture and treatment were the same as described in Section 4.7. After culture, the cells were digested by pancreatin and then centrifuged at 113× *g* for 5 min to collect the supernatant. The GSH content and activities of GCL, GS, GSH-Px, and GR were determined using commercial kits (Nanjing Jiancheng Biotechnology Co., Ltd., Nanjing, China).

### 4.9. Determination of Antioxidative Enzyme Activities in H_2_O_2_-Induced HepG2 Cells

In this study, the activities of superoxide dismutase (SOD), catalase (CAT), NAD(P)H: quinone oxidoreductase 1 (NQO1), and heme oxygenase-1 (HO-1) were determined to explore the antioxidant ability of DH fractions. The HepG2 cells (1 × 10^5^ cells/mL, 200 μL) were seeded into a 96-well plate for 24 h incubation. The methods for cell culture and treatment were the same as described in Section 4.7. The cells were added into 1.0 mL pre-cold PBS and disrupted by an ultrasonic cell disrupter. After centrifugation at 3000× *g* for 10 min, the cell supernatant was collected for the investigation of antioxidative enzymes (SOD, CAT, NQO1, and HO-1) according to the instructions of commercial kits (Nanjing Jiancheng Biotechnology Co., Ltd., Nanjing, China).

### 4.10. Western Blotting Analysis

The proteins from cell samples were extracted by RIPA lysis buffer (Beyotime Biotechnology, Shanghai, China) containing 1% protease inhibitor. After centrifugation (12,000× *g*, 4 °C) for 15 min, the supernatant of proteins was quantified by BCA kits (Beyotime Biotechnology). After boiling for 10 min, the denatured proteins were separated by 10% SDS-PAGE and then transformed onto the polyvinylidene difluoride (PVDF) membrane. The membrane was blocked by 5% defatted milk dissolved in TBST (TBS with 0.1% Tween-20) for 1 h and incubated with Nrf2 (Sevier, Wuhan, China) overnight at 4 °C. After washing with TBST, the membrane was incubated with a second antibody (Abclone, Wuhan, China). The visualization of the protein bands was performed with ECL reagent (Beyotime Biotechnology, Shanghai, China) using an Image-Pro Plus software (LI-COR2800, New Jersey, USA).

### 4.11. Analysis of the Relationship between Chemical Constituents and Bioactivities

The heat map analysis was drawn using the website (http://www.bioinformatics.com.cn/plot_basic_matrix_heatmap_064, accessed on 20 July 2023), and the CF value was used as a comparison with a value of 1 for the different group values of the TFC, TPC, and IC_50_ values of ABTS and DPPH. The bioactivities of various chemical constituents in the DH fractions (DF, EF, and 60%, 80%, 100% fractions) were explored using the Collection of Open Natural Products (COCONUT) database (https://coconut.naturalproducts.net/, accessed on 12 April 2023). The chemical component analysis in different polar fractions of DH and the network diagram of these chemical constituents and their bioactivities were also performed in this study.

### 4.12. Data Analysis

Data was expressed as mean ± standard deviation (SD). The number of samples for testing was six (*n* = 6). Statistical analysis was performed using one-way analysis of variance (ANOVA) followed by a *t*-test to determine the significance of differences (*p* < 0.05). All analyses were performed using Origin 8.5 software (OriginLab, Northampton, MA, USA).

## 5. Conclusions

In this study, 11 polar fractions from DH ethanol extract were obtained. The results showed that various polar fractions of DH extract had different chemical compositions and propositions. Among these fractions DF, EF, and 60%, 80%, and 100% ethanol-aqueous fractions showed higher contents of TFC and TPC and exhibited significant scavenging abilities on ABTS and DPPH radicals. Moreover, these five DH fractions suppressed oxidative stress by increasing SOD, CAT, HO-1, and NQO1 activities; by promoting GSH synthesis and metabolism via the enhancement of GCL, GS, GR, and GSH-Px activities; and by reducing accumulations of ROS and MDA in H_2_O_2_-induced HepG2 cells. The potential antioxidative mechanism may be related to the activation of the Nrf2 pathway. This study verified that various extraction and purification techniques were the important factors for the bioactivities of DH, providing a new insight for the targeted development of DH bioactive constituents.

## Figures and Tables

**Figure 1 ijms-24-12646-f001:**
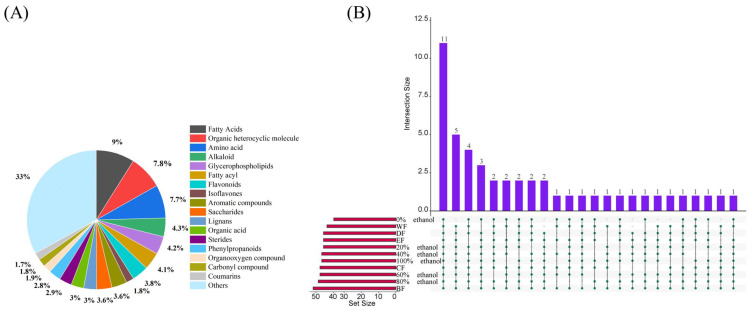
Classifications of DH extracts from different extracted fractions. (**A**) The compound types identified in 11 fractions from DH; (**B**) The UpSet diagram of 11 fractions from DH. DF: Dichloromethane fraction; EF: ethyl acetate fraction; BF: n-butanol fraction; WF: water fraction; CF: crude fraction; 0~100%: 0%, 20%, 40%, 60%, 80%, or 100% ethanol macroporous absorption resin. Each dop represents a chemical compound.

**Figure 2 ijms-24-12646-f002:**
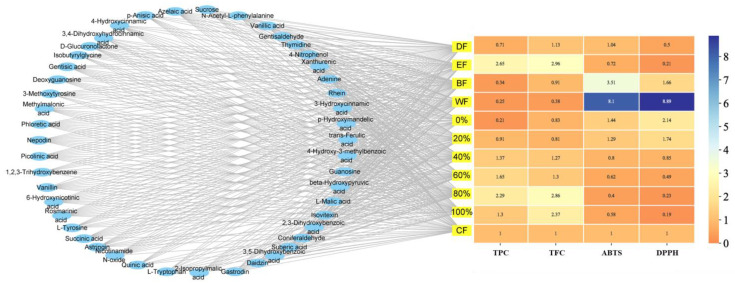
A graph for TPC and TFC correlation with antiradical assay and chemical compositions. TPC: Total phenolic content; TFC: total flavone content; ABTS: 2,2′-azinobis (3-ethylbenzothiazoline-6-sulfonic acid) diammonium salt; DPPH: 1,1-diphenyl-2-picrylhydrazyl; DF: dichloromethane fraction; EF: ethyl acetate fraction; BF: n-butanol fraction; WF: water fraction; CF: crude fraction; 0~100%: 0%, 20%, 40%, 60%, 80%, or 100% ethanol-aqueous macroporous absorption resin. For TPC and TFC lines, the darker the color, the higher the levels of TPC and TFC. For ABTS and DPPH lines, the lighter the color, the lower the IC_50_ values of ABTS and DPPH.

**Figure 3 ijms-24-12646-f003:**
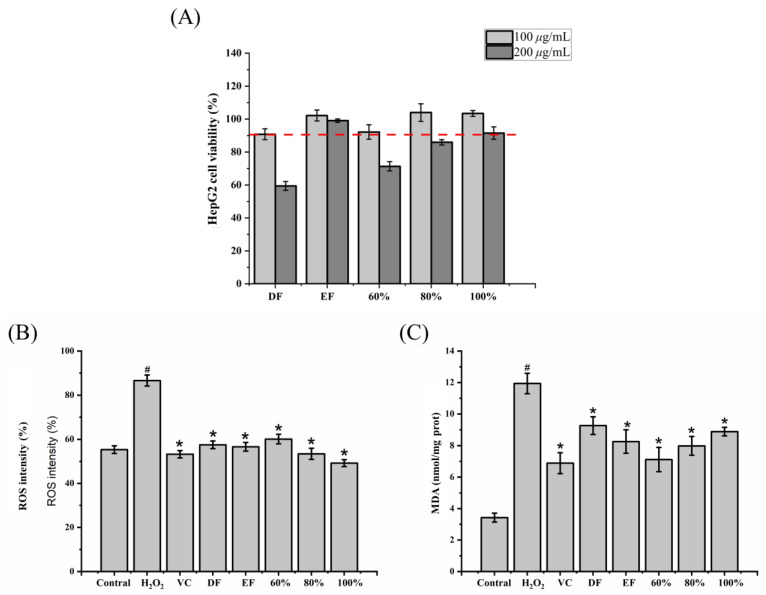
The effect of DH fractions on the viability of HepG2 cells and ROS and MDA accumulation in H_2_O_2_-induced HepG2 cells. (**A**) The viability of HepG2 cells; (**B**) intracellular ROS content; (**C**) MDA content. ROS: Reactive oxygen species; MDA: malondialdehyde; DF: dichloromethane fraction; EF: ethyl acetate fraction; 60~100%: 60%, 80%, or 100% ethanol macroporous absorption resin; VC: vitamin C. “#” represents significance between control group and H_2_O_2_ group (*p* < 0.05); “*” represents significance between H_2_O_2_ and sample groups (*p* < 0.05). The red dotted line in (**A**) represents that the cell viability was more than 90%. The error bar represents a standard deviation.

**Figure 4 ijms-24-12646-f004:**
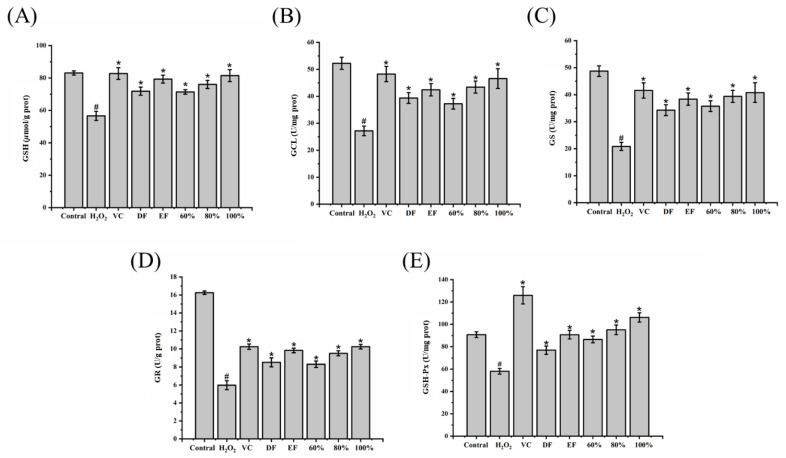
Effects of DH fractions on GSH synthesis and metabolism in H_2_O_2_-induced HepG2 cells. (**A**) GSH content; (**B**) GCL activity; (**C**) GS activity; (**D**) GR activity; (**E**) GSH-Px activity. GSH: Glutathione; GCL: glutamate cysteine ligase; GS: glutathione synthase; GR: glutathione reductase; GSH-Px: glutathione peroxidase; DF: dichloromethane fraction; EF: ethyl acetate fraction; 60~100%: 60%, 80%, or 100% ethanol macroporous absorption resin; VC: vitamin C. “#” represents significance between control group and H_2_O_2_ group (*p* < 0.05); “*” represents significance between H_2_O_2_ and sample groups (*p* < 0.05). The error bar represents the standard deviation (S.D.).

**Figure 5 ijms-24-12646-f005:**
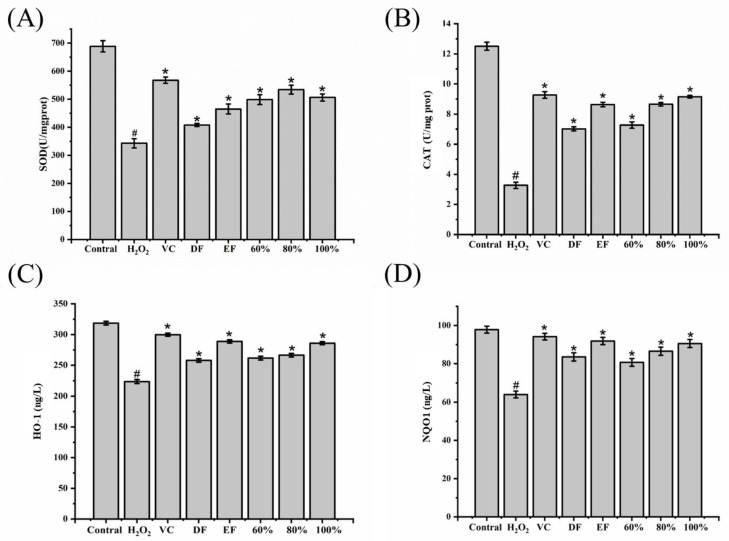
Effects of DH fractions on the activities of antioxidative enzymes in H_2_O_2_-induced HepG2 cells. (**A**) SOD activity; (**B**) CAT activity; (**C**) HO-1 activity; (**D**) NQO1 activity. SOD: Superoxide dismutase; CAT: catalase; HO-1: heme oxygenase-1; NQO1: NAD(P)H: quinone oxidoreductase 1; DF: dichloromethane fraction; EF: ethyl acetate fraction; 60~100%: 60%, 80%, or 100% ethanol macroporous absorption resin; VC: vitamin C. “#” represents significance between control group and H_2_O_2_ group (*p* < 0.05); “*” represents significance between H_2_O_2_ and sample groups (*p* < 0.05). The error bar represents the standard deviation (S.D.).

**Figure 6 ijms-24-12646-f006:**
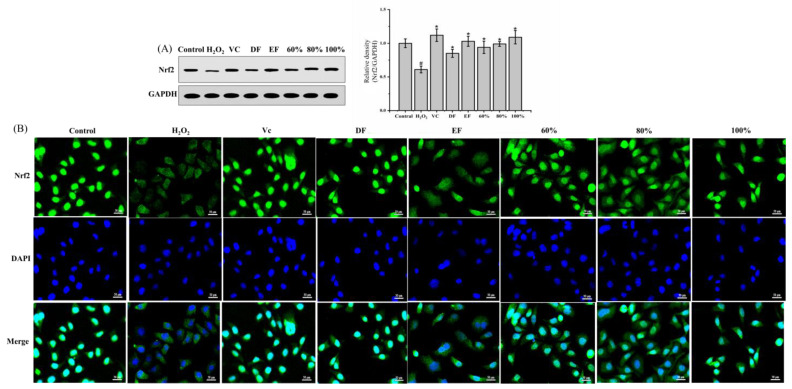
Effects of DH fractions on activities of antioxidative enzymes and Nrf2 expression in H_2_O_2_-induced HepG2 cells. (**A**) Nrf2 protein expressions by Western blotting analysis; (**B**) immunofluorescence results of Nrf2. Nrf2: Nuclear factor erythroid 2-rrelated factor 2; DF: dichloromethane fraction; EF: ethyl acetate fraction; 60~100%: 60%, 80%, or 100% ethanol macroporous absorption resin; VC: vitamin C. “#” represents significance between control group and H_2_O_2_ group (*p* < 0.05); “*” represents significance between H_2_O_2_ and sample groups (*p* < 0.05).

**Figure 7 ijms-24-12646-f007:**
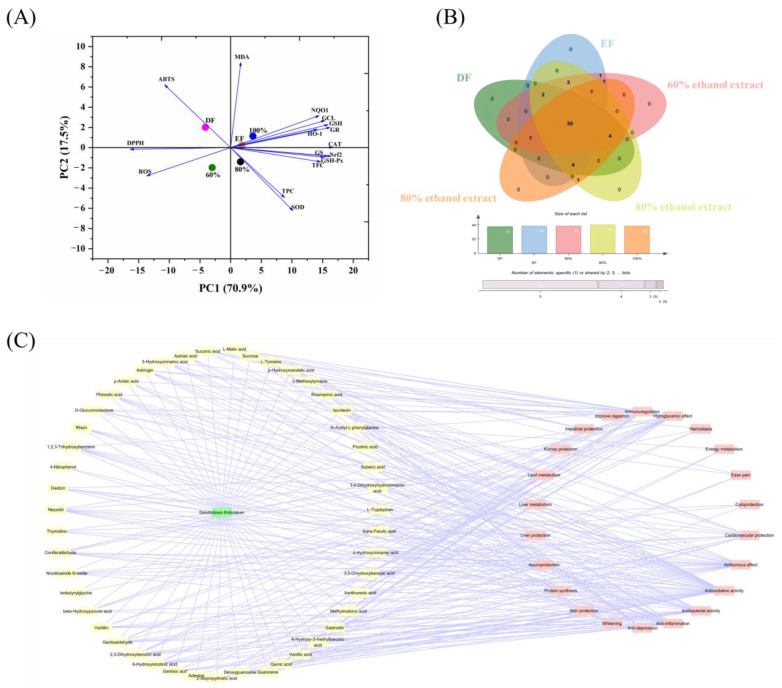
Multivariate analysis and pharmacologic compound network analysis of DH fractions on antioxidative indexes or bioactivities. (**A**) Results of principal component analysis (PCA); (**B**) Venn diagram of DH fractions; (**C**) network between chemical constituents of DH fractions and bioactivities. PCA: principal component analysis; DF: dichloromethane fraction; EF: ethyl acetate fraction; BF: n-butanol fraction; WF: water fraction; CF: crude fraction; 0~100%: 0%, 20%, 40%, 60%, 80%, or 100% ethanol-aqueous macroporous absorption resin.

**Figure 8 ijms-24-12646-f008:**
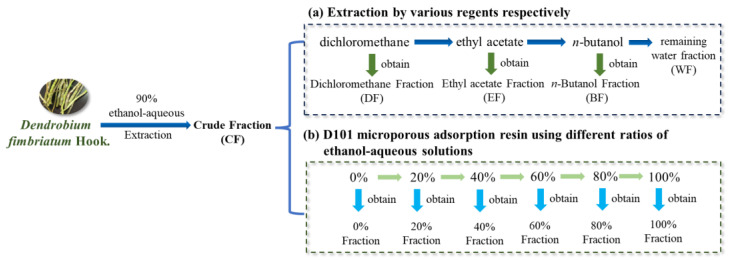
The process flow diagram of *Dendrobium fimbriatum* Hook using different extraction and purification methods. DF: Dichloromethane fraction; EF: ethyl acetate fraction; BF: n-butanol fraction; WF: water fraction; CF: crude fraction; 0~100%: 0%, 20%, 40%, 60%, 80%, or 100% ethanol-aqueous macroporous absorption resin fractions.

**Table 1 ijms-24-12646-t001:** Main compounds identified in different extracts in *Dendrobium fimbriatum* Hook using UPLC-Q-Exactive Orbitrap-MS/MS in negative ion mode (relative quantitation).

	Compounds	RT (s)	[M-H]^−^ (*m*/*z*)	Molecular Formula	DH Extracts										
					CF	DF	EF	BF	WF	0%	20%	40%	60%	80%	100%
1	*p*-Anisic acid	2.288025	151.0404394	C_8_H_8_O_3_	0.0916	0.2001	0.0771	3.4933	——	——	0.2794	0.1426	0.1416	0.0326	0.0329
2	Phloretic acid	17.24835	165.0560322	C_9_H_10_O_3_	0.0350	0.0271	0.0397	0.0007	0.1344	0.0591	0.2124	0.1057	0.0288	0.0125	0.0084
3	D-Glucuronolactone	38.043	221.0302291	C_6_H_8_O_6_	0.0891	0.0031	0.2907	3.4858	0.1515	0.0732	——	0.1314	0.0039	0.0253	0.0029
4	Rhein	42.3867	283.0286209	C_15_H_8_O_6_	0.0228	——	0.2543	7.9278	0.0104	0.2289	——	0.1162	0.0372	——	——
5	1,2,3-Trihydroxybenzene	46.63125	171.0288259	C_6_H_6_O_3_	0.0738	0.7312	0.0860	6.2935	0.0040	——	0.0093	0.3319	0.0965	0.1797	——
6	4-Nitrophenol	49.12335	138.0197092	C_6_H_5_NO_3_	0.5999	0.3267	0.4740	16.8797	——	——	0.4759	0.4276	0.4833	0.3024	0.2754
7	Daidzin	49.8962	415.1091951	C_21_H_20_O_9_	0.0753	0.0646	0.0626	13.7131	——	0.0961	0.4082	0.0707	0.0262	0.0431	0.0079
8	Nepodin	69.69	215.0738272	C_13_H_12_O_3_	0.2885	0.0421	0.7701	18.2224	0.2139	0.2776	0.2865	0.2557	0.0230	0.0535	0.0206
9	Thymidine	75.919	241.0875386	C_10_H_14_N_2_O_5_	0.1239	0.5287	——	7.0666	0.0320	0.0376	0.0463	0.3986	0.0701	0.0514	0.0189
10	Coniferaldehyde	79.23495	177.0557204	C_10_H_10_O_3_	0.2182	0.6817	0.1459	0.0007	0.0000	0.1593	1.1408	0.6181	0.0640	1.6804	0.4695
11	Nicotinamide N-oxide	87.9813	137.0357308	C_6_H_6_N_2_O_2_	2.8887	2.3539	0.3391	2.8195	0.2324	0.4231	0.4566	0.8607	0.7632	0.2891	——
12	Isobutyrylglycine	91.5977	144.0668483	C_6_H_11_NO_3_	0.2093	0.0284	0.8858	0.0007	0.0031	0.0176	——	——	0.2785	0.1729	0.0230
13	β-Hydroxypyruvic acid	94.436	103.0037513	C₃H₄O₄	——	0.1528	0.0628	0.0007	——	——	0.0217	0.0687	0.0672	0.1523	0.0266
14	Vanillin	98.6035	151.0404444	C_8_H_8_O_3_	1.4805	2.5335	——	0.0007	0.6991	0.8441	2.3695	0.5707	0.1337	0.8053	0.1778
15	Gentisaldehyde	114.6265	137.0246661	C_7_H_6_O_3_	0.2238	0.2754	0.3780	4.9913	0.1568	0.2609	0.3035	0.3042	——	0.3003	0.2612
16	2,3-Dihydroxybenzoic acid	132.969	153.0196768	C_7_H_6_O_4_	0.0391	0.0528	0.0362	4.8839	0.0180	——	0.0456	0.0029	0.0004	0.1296	0.0029
17	6-Hydroxynicotinic acid	147.72	138.0198821	C_6_H_5_NO_3_	0.2631	0.1204	0.1961	0.0007	0.1459	0.1205	0.1785	0.0184	0.1979	0.0667	0.0816
18	Gentisic acid	160.302	153.0196684	C_7_H_6_O_4_	0.2443	0.1166	0.1670	0.0007	——	0.1037	0.1672	0.0475	0.0360	0.0525	0.0737
19	Adenine	168.86	134.0472648	C_5_H_5_N_5_	0.0834	——	0.0037	1.3238	0.0235	0.0139	0.0162	0.0204	0.0262	0.0438	0.1442
20	2-Isopropylmalic acid	171.5385	175.0614802	C_7_H_12_O_5_	0.0269	——	0.1579	0.1507	0.0188	0.0152	0.0231	0.0225	0.0226	0.1158	——
21	Deoxyguanosine	181.783	266.0895776	C_10_H_13_N_5_O_4_	0.3232	0.0936	0.2640	8.7179	0.2685	——	0.4353	0.0949	0.1397	0.1742	0.1062
22	Guanosine	188.495	282.0850112	C_10_H_13_N_5_O_5_	0.9932	0.0669	1.5511	5.8745	0.3280	——	0.0709	0.0646	0.1094	0.0712	0.0696
23	Quinic acid	189.151	191.0564597	C_7_H_12_O_6_	4.9520	0.4802	0.4181	52.3224	5.6111	6.6556	——	——	——	0.1459	0.7790
24	Vanillic acid	189.359	167.0353101	C_8_H_8_O_4_	0.2242	0.1470	0.1225	6.6700	0.0798	0.1173	0.2849	0.1038	0.0382	0.0666	0.1925
25	4-Hydroxy-3-methylbenzoic acid	190.0785	151.0404309	C_8_H_8_O_3_	0.0144	0.0099	0.0249	0.2518	0.0142	0.0169	0.0638	0.0203	0.0362	0.0428	0.0044
26	Gastrodin	196.616	345.118909	C_13_H_18_O_7_	0.1607	0.1935	0.1864	0.0007	0.0549	——	0.2060	0.0791	0.0482	0.0882	0.1636
27	Methylmalonic acid	204.8075	117.0195125	C_4_H_6_O_4_	5.7798	0.8711	——	14.3993	0.0786	——	3.1999	15.0147	8.7629	0.7702	0.1527
28	Xanthurenic acid	208.4115	204.0302746	C_10_H_7_NO_4_	0.6953	0.0734	0.1787	17.1292	0.0000	0.1620	0.1069	0.0682	0.1043	0.0114	0.0769
29	3,5-Dihydroxybenzoic acid	216.7765	153.0196894	C_7_H_6_O_4_	——	0.1626	1.9679	39.9793	2.6082	——	2.0348	1.6361	0.8824	0.9116	0.2098
30	4-Hydroxycinnamic acid	219.943	163.0403603	C_9_H_8_O_3_	0.0636	0.0129	0.1153	0.0007	——	0.0373	0.0934	0.0356	0.0336	0.0412	0.0238
31	trans-ferulic acid	224.396	193.0506035	C_10_H_10_O_4_	0.3113	0.0612	——	0.7106	0.0017	0.0000	0.0585	0.1160	0.9230	0.5970	0.0647
32	L-Tryptophan	225.028	203.0828503	C_11_H_12_N_2_O_2_	0.0708	0.0547	0.0509	0.5656	——	0.0139	0.0729	——	0.0525	0.0227	0.0044
33	3,4-Dihydroxyhydrocinnamic acid	225.145	181.0508496	C_9_H_10_O_4_	0.0060	——	——	0.0007	0.0172	——	0.0115	0.0168	——	0.0551	0.0118
34	Suberic acid	240.8435	173.0822936	C_8_H_14_O_4_	0.0088	0.0907	0.1320	0.0007	0.0000	0.0593	0.0103	——	0.0261	0.1225	0.0821
35	Picolinic acid	242.2815	122.0247855	C_6_H_5_NO_2_	0.1511	0.3593	0.2585	1.0082	0.0000	0.1341	0.1935	0.1655	0.2526	0.2742	0.1714
36	N-Acetyl-L-phenylalanine	248.142	206.0822059	C_11_H_13_NO_3_	0.0572	0.1233	0.1603	6.3544	0.0000	0.0494	0.1125	0.0668	0.1455	——	0.1045
37	Isovitexin	261.957	431.1011504	C_21_H_20_O_10_	2.9303	2.8189	3.2317	42.3072	1.2943	——	3.5981	1.3451	0.8434	1.7325	2.5823
38	Rosmarinic acid	265.2535	359.0776065	C_18_H_16_O_8_	1.9446	1.1495	4.8024	5.1753	0.2191	0.8922	1.0642	0.8161	2.8470	1.5225	0.3295
39	3-Methoxytyrosine	267.141	210.0772445	C_10_H_13_NO_4_	1.9539	0.5663	1.8479	8.5020	0.1042	0.0850	0.3534	2.1029	——	2.2729	0.8262
40	p-Hydroxymandelic acid	270.556	167.0353064	C_8_H_8_O_4_	2.0663	——	2.5993	8.8652	0.0370	0.1083	0.4502	0.1708	0.7354	——	2.9926
41	L-Tyrosine	274.5095	180.0668782	C_9_H_11_NO_3_	——	0.0858	1.7178	13.3174	0.7696	0.8725	——	1.9643	0.8848	0.7403	0.1039
42	Sucrose	274.651	341.1099464	C_12_H_22_O_11_	0.1667	0.4042	0.2833	6.5297	0.1725	0.2127	——	0.3963	0.0753	0.1684	0.2092
43	L-Malic acid	290.951	133.0143139	C_4_H_6_O_5_	0.0370	0.0369	——	0.0007	0.0861	0.0091	0.0907	0.1002	0.0021	0.0052	0.0015
44	Succinic acid	293.207	117.0194885	C_4_H_6_O_4_	5.7112	——	11.0985	5.3234	0.5606	0.3551	0.4933	0.6221	0.7346	4.8614	——
45	Azelaic acid	304.671	187.0976293	C_9_H_16_O_4_	0.0965	0.0485	0.0751	3.0576	0.0289	0.0349	0.0899	0.0438	0.0891	0.0445	0.0511
46	3-Hydroxycinnamic acid	307.7435	163.0404304	C_18_H_14_O_5_	——	0.2008	0.0240	0.0007	——	0.0880	0.0763	0.0491	0.0066	0.0890	0.1737
47	Astringin	310.984	405.1199949	C_20_H_22_O_9_	0.4726	0.2142	0.9561	10.5820	0.0307	0.1411	0.2567	——	0.3422	0.3105	0.0212

DF: Dichloromethane fraction; EF: ethyl acetate fraction; BF: n-butanol fraction; WF: water fraction; CF: crude fraction; 0~100%: 0%, 20%, 40%, 60%, 80%, or 100% ethanol macroporous absorption resin.

**Table 2 ijms-24-12646-t002:** Contents of phenolic content (TPC) and total flavone content (TFC) in *Dendrobium fimbriatum* fractions.

Sample	TPC (mg/g)	TFC (mg/g)
DF	79.27 ± 2.39 ^h^	17.91 ± 1.30 ^e^
EF	298.13 ± 1.59 ^a^	47.05 ± 1.73 ^a^
BF	38.20 ± 1.77 ^i^	14.51 ± 0.43 ^g^
WF	27.93 ± 0.60 ^j^	6.05 ± 1.04 ^h^
0%	23.32 ± 2.39 ^k^	13.11 ± 1.87 ^g^
20%	102.62 ± 0.40 ^g^	12.93 ± 0.94 ^g^
40%	153.54 ± 2.78 ^d^	20.15 ± 0.61 ^d^
60%	185.04 ± 1.59 ^c^	20.68 ± 0.35 ^d^
80%	256.77 ± 2.16 ^b^	45.44 ± 1.54 ^b^
100%	145.94 ± 0.01 ^e^	37.67 ± 1.54 ^c^
CF	112.32 ± 1.39 ^f^	15.88 ± 0.35 ^f^

Different letters represent that the statistical difference is significant. *p* < 0.05. DF: Dichloromethane fraction; EF: ethyl acetate fraction; BF: n-butanol fraction; WF: water fraction; CF: crude fraction; 0~100%: 0%, 20%, 40%, 60%, 80%, or 100% ethanol macroporous absorption resin. The different letters represent the standard deviation (S.D.)

**Table 3 ijms-24-12646-t003:** The ABTS and DPPH free radical scavenging ability and IC_50_ values in various polar fractions of *Dendrobium fimbriatum*.

Free Radical	Fractions	Concentrations (μg/mL)					IC_50_ Values (μg/mL)
		1000	500	250	125	62.5	
ABTS free radical	DF	94.28 ± 1.26	83.01 ± 1.80	55.85 ± 0.96	39.61 ± 0.84	27.51 ± 0.29	158.87 ± 2.20 ^e^
	EF	152.15 ± 1.42	98.57 ± 1.69	88.98 ± 1.69	61.43 ± 1.33	26.45 ± 1.95	109.45 ± 2.04 ^h^
	BF	59.58 ± 2.41	51.77 ± 1.67	35.69 ± 2.34	24.96 ± 1.61	18.10 ± 0.72	534.70 ± 2.73 ^b^
	WF	30.47 ± 2.61	21.81 ± 1.05	16.19 ± 0.86	12.98 ± 1.07	9.89 ± 1.22	1234.88 ± 3.09 ^a^
	0%	84.21 ± 2.26	74.93 ± 1.68	50.93 ± 1.03	31.68 ± 1.28	22.13 ± 0.82	220.08 ± 2.34 ^c^
	20%	84.74 ± 2.28	78.51 ± 1.19	50.45 ± 1.38	34.93 ± 2.28	22.40 ± 0.58	196.22 ± 2.29 ^d^
	40%	92.08 ± 1.24	89.71 ± 2.93	68.92 ± 1.93	46.70 ± 2.08	33.33 ± 2.26	122.45 ± 2.09 ^g^
	60%	97.55 ± 1.52	96.47 ± 1.86	79.47 ± 0.54	56.02 ± 2.47	37.79 ± 1.87	94.68 ± 1.98 ^i^
	80%	99.67 ± 0.52	99.39 ± 1.50	94.88 ± 1.80	62.29 ± 0.67	42.94 ± 2.03	60.49 ± 1.78 ^k^
	100%	100.17 ± 2.33	99.50 ± 0.53	84.38 ± 1.90	56.17 ± 0.60	35.77 ± 1.35	88.33 ± 1.92 ^j^
	CF	74.72 ± 0.33	72.76 ± 2.35	70.34 ± 1.11	44.30 ± 1.18	29.13 ± 2.05	152.45 ± 2.18 ^f^
	Vc	—	—	—	99.77 ± 0.29	85.05 ± 1.38	21.21 ± 1.33 ^l^
DPPH free radical	DF	88.22 ± 1.81	70.18 ± 0.60	52.44 ± 1.99	33.65 ± 1.98	25.00 ± 1.17	208.35 ± 2.32 ^g^
	EF	95.55 ± 0.39	93.98 ± 0.56	82.64 ± 2.70	55.14 ± 1.84	42.48 ± 1.89	87.21 ± 1.94 ^i^
	BF	54.43 ± 1.39	43.16 ± 1.73	33.26 ± 0.92	26.25 ± 1.26	18.32 ± 0.99	678.18 ± 2.83 ^d^
	WF	22.11 ± 0.42	16.32 ± 1.39	11.77 ± 1.89	9.33 ± 0.98	7.61 ± 1.20	3681.58 ± 3.57 ^a^
	0%	48.66 ± 1.13	34.30 ± 0.14	23.02 ± 0.05	16.63 ± 0.14	13.37 ± 0.28	885.66 ± 2.95 ^b^
	20%	54.29 ± 1.55	45.58 ± 2.92	32.34 ± 0.56	21.09 ± 2.15	14.84 ± 1.01	721.35 ± 2.86 ^c^
	40%	75.57 ± 2.04	54.75 ± 1.60	40.28 ± 1.23	27.77 ± 1.28	19.65 ± 0.71	350.94 ± 2.55 ^f^
	60%	87.30 ± 2.15	72.23 ± 1.42	52.59 ± 1.31	35.10 ± 0.49	24.1 ± 1.67	204.95 ± 2.31 ^g^
	80%	93.17 ± 0.36	89.16 ± 0.72	78.25 ± 0.31	57.63 ± 1.12	37.68 ± 2.01	93.67 ± 1.97 ^h^
	100%	93.24 ± 0.31	89.36 ± 0.20	79.99 ± 0.81	63.25 ± 1.09	42.44 ± 1.28	78.43 ± 1.89 ^j^
	CF	69.00 ± 2.36	56.55 ± 1.01	36.90 ± 1.09	23.15 ± 1.05	15.82 ± 1.83	414.33 ± 2.62 ^e^
	Vc	—	—	—	98.36 ± 1.59	92.03 ± 1.26	17.54 ± 1.24 ^k^

Note: Different letters represent that statistical difference is significant. *p* < 0.05. DF: Dichloromethane fraction; EF: ethyl acetate fraction; BF: n-butanol fraction; WF: water fraction; CF: crude fraction; 0~100%: 0%, 20%, 40%, 60%, 80%, or 100% ethanol-aqueous macroporous absorption resin; Vc: vitamin C.

## Data Availability

The data are contained within the article.

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
