# Peer review of "A Comparative Analysis of Chemical Constituents and Antioxidant Effects of *Dendrobium fimbriatum* Hook Fractions with Different Polarities"

_ijms, 2023, doi:10.3390/ijms241612646_

Round 1

Reviewer 1 Report

The work is rich in data and very interesting, but I have some points to make:

1) Please check the refercences number in the text because there are same mistakes as at the beginning of section 2;

2) Figure 2 is not completely clear about the data;

3) At line 102, it should be 1.04

4) A graph for TPC and TFC correlation with antiradical assay and mass spectrometry results should be added to better clarify the most promising fractions. 

5) I suggest to improve the discussion section 

6) In the section Materials and Methods the data about mass spectrometry parameters are not reported. Please add this data. 

7) In the section of antioxidant activity the concentrations about fractions should be added. 

The English should be revised. 

Author Response

Dear Editors and Reviewers:

Thanks for your supports and suggestions, and we have revised our manuscript according to the reviewers’ comments and responded these questions carefully. We hope the reversion may meet the requirements of your esteem journal and reviewers.

The work is rich in data and very interesting, but I have some points to make:

1) Please check the references number in the text because there are same mistakes as at the beginning of section 2;

Response: Thanks for pointing out our mistakes, and we carefully rechecked all the references in the manuscript and revised the wrong numbers.

 2) Figure 2 is not completely clear about the data;

Response: Thanks for your suggestion, we replaced the clearer image of Figure 2, and we also provided the original image as attachments to the system of IJMS.

3) At line 102, it should be 1.04

Response: Thanks for pointing out our mistakes, and we changed the “6.05 ± 2.04” to “6.05 ± 1.04”.

4) A graph for TPC and TFC correlation with antiradical assay and mass spectrometry results should be added to better clarify the most promising fractions.

Response: Thanks for your valuable suggestion, we tried to construct a graph for TPC and TFC correlation with antiradical assay and mass spectrometry results, and added this image in the manuscript (Figure 2) as follows: see the attachment or manuscript.

5) I suggest to improve the discussion section.

Response: Thanks for your suggestion, and we improved the discussion in the manuscript, see red sections.

6) In the section Materials and Methods the data about mass spectrometry parameters are not reported. Please add this data.

Response: Thanks for your suggestion, and we supplemented the data about mass spectrometry parameters in the section of “Chemical constituent analysis of DH fractions” as follows:

The key parameters of the mass spectrometry were as follows: the spray voltage was +3.8 and −3.4 kV; the sheath gas flow rate was 50 arbitrary units (arb); the auxiliary gas flow rate was 15 units (arb); the capillary temperature was 320°C and the auxiliary gas heater was 350°C; the scanning mode was the full scan with a 60000 resolution, and the resolution of data-related MS/MS was 15000; the charge transfer and normalized collision energies were10, 30 and 60eV, respectively. The data acquisition and processing were carried out using the software Xcalibur 4.1 (Waltham, MA, Thermo Scientific).

7) In the section of antioxidant activity, the concentrations about fractions should be added.

Response: Thanks for your suggestion, the concentrations for determining the scavenging abilities of ABTS and DPPH were 62.5, 125, 250, 500, 1000 μg/mL, and then the IC50 values were calculated. The specific data is as follows, and relative data was also supplemented in Table 3 in the manuscript as follows .

Table 3 The ABTS and DPPH free radical scavenging ability and IC50 values in various polar fractions of Dendrobium fimbriatum Hook

Free radical

Fractions

Concentrations (μg/mL)

IC50 values

(μg/mL)

1000

500

250

125

62.5

ABTS free radical

DF

94.28±1.26

83.01±1.80

55.85±0.96

39.61±0.84

27.51±0.29

158.87±2.20e

EF

152.15±1.42

98.57±1.69

88.98±1.69

61.43±1.33

26.45±1.95

109.45±2.04h

BF

59.58±2.41

51.77±1.67

35.69±2.34

24.96±1.61

18.10±0.72

534.70±2.73b

WF

30.47±2.61

21.81±1.05

16.19±0.86

12.98±1.07

9.89±1.22

1234.88±3.09a

0%

84.21±2.26

74.93±1.68

50.93±1.03

31.68±1.28

22.13±0.82

220.08±2.34c

20%

84.74±2.28

78.51±1.19

50.45±1.38

34.93±2.28

22.40±0.58

196.22±2.29d

40%

92.08±1.24

89.71±2.93

68.92±1.93

46.70±2.08

33.33±2.26

122.45±2.09g

60%

97.55±1.52

96.47±1.86

79.47±0.54

56.02±2.47

37.79±1.87

94.68±1.98i

80%

99.67±0.52

99.39±1.50

94.88±1.80

62.29±0.67

42.94±2.03

60.49±1.78k

100%

100.17±2.33

99.50±0.53

84.38±1.90

56.17±0.60

35.77±1.35

88.33±1.92j

CF

74.72±0.33

72.76±2.35

70.34±1.11

44.30±1.18

29.13±2.05

152.45±2.18f

Vc

99.77±0.29

85.05±1.38

21.21±1.33l

DPPH free radical

DF

88.22±1.81

70.18±0.60

52.44±1.99

33.65±1.98

25.00±1.17

208.35±2.32g

EF

95.55±0.39

93.98±0.56

82.64±2.70

55.14±1.84

42.48±1.89

87.21±1.94i

BF

54.43±1.39

43.16±1.73

33.26±0.92

26.25±1.26

18.32±0.99

678.18±2.83d

WF

22.11±0.42

16.32±1.39

11.77±1.89

9.33±0.98

7.61±1.20

3681.58±3.57a

0%

48.66±1.13

34.30±0.14

23.02±0.05

16.63±0.14

13.37±0.28

885.66±2.95b

20%

54.29±1.55

45.58±2.92

32.34±0.56

21.09±2.15

14.84±1.01

721.35±2.86c

40%

75.57±2.04

54.75±1.60

40.28±1.23

27.77±1.28

19.65±0.71

350.94±2.55f

60%

87.30±2.15

72.23±1.42

52.59±1.31

35.10±0.49

24.1±1.67

204.95±2.31g

80%

93.17±0.36

89.16±0.72

78.25±0.31

57.63±1.12

37.68±2.01

93.67±1.97h

100%

93.24±0.31

89.36±0.20

79.99±0.81

63.25±1.09

42.44±1.28

78.43±1.89j

CF

69.00±2.36

56.55±1.01

36.90±1.09

23.15±1.05

15.82±1.83

414.33±2.62e

Vc

98.36±1.59

92.03±1.26

17.54±1.24k

8)The English should be revised.

Response: Thanks for your suggestion, we have revised language by a native English speaker.

Reviewer 2 Report

The article titled: A Comparative Analysis of Chemical Constituents and Antioxidant Effects of Dendrobium fimbriatum Hook Fractions with Different Polarities, presents the chemical and biological characterization of Dendrobium fimbriatum extract’s fractions obtained using different solvents and macroporous resins. The manuscript needs to be verified and corrected before publication. Below I present my remarques.

- the abbreviation CE used in the abstract needs to be explained - is it the same as the one used in the CF tables? I need clarification on this. The abbreviation CE does not appear in the tables.

- I do not understand the term "enrichment technique" concerning the "extraction liquid-liquid" and "macroporous resins purification techniques". The term "enrichment technique" is mentioned many times. In my opinion, it does not reflect how the authors dealt with the crude extract. I am asking for an explanation that will convince me of the correct use of this term.

- y-axis captions in Figure 1B are not visible (too small)

- In the Table 1 header is: Dendrobium Fimbriatum Hook when it should be: Dendrobium fimbriatum Hook

- such a conclusion: "These findings suggested that different extracted methods could affect the contents of TPC and TFC in DH fractions" – is incorrect. One extraction method (it also demands the additions - temperature and time of extraction) was used, later the extract was only fractionated using different solvents or by separation on macroporous resins. These activities cannot be called "different extraction methods." It is self-explanatory that treating the base extract with different reagents will result in fractions rich in active compounds in different ways.

- statement: “The results (Table 3) showed that the ABTS and DPPH scavenging abilities were positively correlated with the concentrations of DH fractions in different solutions.” is imprecise because this Table does not contain TPC and TFC results; therefore, it cannot lead to such a conclusion.

- there is no description of how the presence of specific substances in the tested DH fractions was found. There are no given standards; with such many detected compounds, their use is not possible. How was the identification done? In addition, even though the plant matrix is rich in compounds, the detection of 2300 in 90% ethanol extract (because this is the crude extract to which the content should be referred) seems improbable to me ... Hence, I am asking for a detailed description of the identification of these substances.

- the designation of the resin fraction uses the abbreviation ethanol-aqueous solutions, respectively, to obtain 0% fraction - that is,  is it a fraction obtained using 100% ethanol?

- captions under figures 2, and 3 must explain the abbreviations used in the graphs

- in TPC - why use incubation at 70 degrees C when most of the available methodologies use room temperature for this determination?

- in different places, the authors take TFC differently: flavonoid content, in another total flavonol content

- paragraph: "4.6. Determination of ROS content in H2O2 induced HepG2 cells"  is not well described; for example, at what concentration were the samples or vitamin C tested? The important information is needed.

- The authors mention a Figure 4V in the text - it does not exist

- in the paragraph: "4.8. Determination of antioxidative enzyme activities in H2O2-induced HepG2 cells", there is practically no information about the conditions of the experiment - what concentrations were tested and how the results were expressed - only short information that kits were used. This point should be better described.

- ABTS and DPPH methodology descriptions need to be completed; there are no concentrations of the tested samples or prepared reagents, as well as the lack of basic parameters of the assay. It needs to be supplemented.

Author Response

Dear Editors and Reviewers:

Thanks for your supports and suggestions, and we have revised our manuscript according to the reviewers’ comments and responded these questions carefully. We hope the reversion may meet the requirements of your esteem journal and reviewers.

Reviewer 2:

The article titled: A Comparative Analysis of Chemical Constituents and Antioxidant Effects of Dendrobium fimbriatum Hook Fractions with Different Polarities, presents the chemical and biological characterization of Dendrobium fimbriatum extract’s fractions obtained using different solvents and macroporous resins. The manuscript needs to be verified and corrected before publication. Below I present my remarques.

- the abbreviation CE used in the abstract needs to be explained - is it the same as the one used in the CF tables? I need clarification on this. The abbreviation CE does not appear in the tables.

Response: Thanks for pointing out our spelling mistakes. The abbreviation “CE” is “CF”, and we revised the mistakes in the abstract.

- I do not understand the term "enrichment technique" concerning the "extraction liquid-liquid" and "macroporous resins purification techniques". The term "enrichment technique" is mentioned many times. In my opinion, it does not reflect how the authors dealt with the crude extract. I am asking for an explanation that will convince me of the correct use of this term.

Response: Thanks for your suggestion, and we quite agree with you that the “enrichment technique” could not be used to represent the "extraction liquid-liquid" and "macroporous resins purification techniques". We have revised this term in the manuscript (see red sections). For a clearer understanding of the enrichment technique, we drew a schematic presentation in the 4.2 section and the image is as follows: (see the attachment or manuscript)

As shown in the schematic diagram, the CF was extracted from DH by the 90% ethanol--aqueous. Then the chemical components of CF were enriched or purification according to the solubleness and polarity using two methods: (a) various regents of dichloromethane, ethyl acetate, n-butanol, respectively; (b) D101 macroporous adsorption resin via different ratios of ethanol-aqueous solutions, respectively. Thereby, different chemical components could be enriched or purification in different fractions.

- y-axis captions in Figure 1B are not visible (too small)

Response: Thanks for your suggestion, we have redrawn the Figure 1B and rearranged the Fig 1 in the manuscript (see Figure 1) as follows:

- In the Table 1 header is: Dendrobium Fimbriatum Hook when it should be: Dendrobium fimbriatum Hook

Response: Thanks for pointing our spelling mistakes, and we have revised “Dendrobium Fimbriatum Hook” to “Dendrobium fimbriatum Hook”

- such a conclusion: "These findings suggested that different extracted methods could affect the contents of TPC and TFC in DH fractions" – is incorrect. One extraction method (it also demands the additions - temperature and time of extraction) was used, later the extract was only fractionated using different solvents or by separation on macroporous resins. These activities cannot be called "different extraction methods." It is self-explanatory that treating the base extract with different reagents will result in fractions rich in active compounds in different ways.

Response: Thanks for your valuable comments, and we agreed with you that it is not inappropriate to use the term of “different extracted methods” in this manuscript. We revised this term by “different enrichment techniques” in the manuscript.

- statement: “The results (Table 3) showed that the ABTS and DPPH scavenging abilities were positively correlated with the concentrations of DH fractions in different solutions.” is imprecise because this Table does not contain TPC and TFC results; therefore, it cannot lead to such a conclusion.

Response: Thanks for your valuable comments, and we revised the description of the results in Table 3 (see red section of 2.3) as follows: “The results (Table 3) showed that various DH fractions had different IC50 values of ABTS and DPPH. Among them, EF, DF, 60 – 100% ethanol-aqueous fractions exhibited better ABTS and DPPH scavenging abilities, which may be related to their high levels of chemical constituents, such as phenols and flavonoids.”

- there is no description of how the presence of specific substances in the tested DH fractions was found. There are no given standards; with such many detected compounds, their use is not possible. How was the identification done? In addition, even though the plant matrix is rich in compounds, the detection of 2300 in 90% ethanol extract (because this is the crude extract to which the content should be referred) seems improbable to me ... Hence, I am asking for a detailed description of the identification of these substances.

Response: In this study, a non-targeted metabolomics approach was performed to analyse the chemical compositions of various DH fractions. The key parameters of the mass spectrometry were supplemented in the manuscript, see 4.3 section. The metabolomics is generally used to determine the small molecular substances (< 1000 KDa) and their contents, including sugars, organic acids, lipids, vitamins, amino acids, and aromatic hydrocarbons. Its technical platforms are mainly divided into the nuclear magnetic resonance (NMR), gas chromatography-mass spectrometry (GC-MS) and liquid chromatography-mass spectrometry (LC-MS) and capillary electrophoresis with mass spectrometry (CE-MS). In comparison with the database, a large amount of metabolite data will be obtained by the systematic and comprehensive analysis of the entire metabolome. To date, non-targeted metabolomics analysis is widely used in many studies (Castro-Alves, V., Kalbina, I., Nilsen, A., Aronsson, M., Rosenqvist, E., Jansen, M.A.K., et al., 2021. Integration of non-target metabolomics and sensory analysis unravels vegetable plant metabolite signatures associated with sensory quality: A case study using dill (Anethum graveolens). Food Chem. 344, 128714; Kumar, S., Sumner, B.W., Sumner, L.W. 7.08 - Modern Plant Metabolomics for the Discovery and Characterization of Natural Products and Their Biosynthetic Genes. 2020: 156-188; Li, B., Liu, K., Kwok, L., Guo, S., Bai, L., Yang, X., et al., 2022. Development of a non-target metabolomics-based screening method for elucidating metabolic and probiotic potential of bifidobacteria. Innov. Food Sci. Emerg. 77, 102971; Li, B., Liu, K., Kwok, L., Guo, S., Bai, L., Yang, X., et al., 2022. Development of a non-target metabolomics-based screening method for elucidating metabolic and probiotic potential of bifidobacteria. Innov. Food Sci. Emerg. 77, 102971). Thereby, a large amount of chemical substances was collected using the non-targeted metabolomics approach for further analysis in this study.

- the designation of the resin fraction uses the abbreviation ethanol-aqueous solutions, respectively, to obtain 0% fraction - that is, is it a fraction obtained using 100% ethanol?

Response: In this study, 0% ethanol-aqueous solutions mean the water without ethanol. Thereby, 0% fraction was the water fraction.

- captions under figures 2, and 3 must explain the abbreviations used in the graphs

Response: Thanks for your suggestion, we supplemented the explanation of the abbreviations used in the graphs, see red sections.

- in TPC - why use incubation at 70 degrees C when most of the available methodologies use room temperature for this determination?

Response: In this study, the determination of TPC was in accordance with our previous studies (Wang, Y., Wang, Z., Xue, Q., Zhen, L., Wang, Y., Cao, J., et al., 2023. Effect of ultra-high pressure pretreatment on the phenolic profiles, antioxidative activity and cytoprotective capacity of different phenolic fractions from Que Zui tea. Food Chem. 409, 135271; Cheng, C., Gu, Q., Zhang, J., Tao, J., Zhao, T., Cao, J., et al., 2022. Phenolic Constituents, Antioxidant and Cytoprotective Activities, Enzyme Inhibition Abilities of Five Fractions from Vaccinium dunalianum Wight. Molecules 27(11), 3432; Fan, Z., Wang, Y., Yang, M., Cao, J., Khan, A., Cheng, G., 2020. UHPLC-ESI-HRMS/MS analysis on phenolic compositions of different E Se tea extracts and their antioxidant and cytoprotective activities. Food Chem. 318, 126512). Moreover, the method for determination of TPC could be traced to a previously reported colorimetric method (Singleton and Rossi, 1965: DOI: 10.5344/ajev.1965.16.3.144). Compared with the room temperature for 60~120 min, the treatment of 70 ℃ can reduce the reaction time to 10 min. Thereby, we selected the 70 ℃ for the reaction temperature.

- in different places, the authors take TFC differently: flavonoid content, in another total flavonol content.

Response: Thanks for your comments, we will unify the term of “total flavonol content” in the manuscript.

- paragraph: "4.6. Determination of ROS content in H2O2 induced HepG2 cells" is not well described; for example, at what concentration were the samples or vitamin C tested? The important information is needed.

Response: Thanks for your comments, we supplemented the detailed information in the section 4.77 as follows:

The measurement of ROS in H2O2 induced HepG2 cells was according to the method described by Zhao with minor modifications (19). HepG2 cells were grown for 12 h with a density of 1×105 cell/mL in the 6 well plate. The sample groups (100 μg/mL of 5 DH fractions, 1 mL), the Vc group (100 μg/mL Vc, 1 mL,) and the H2O2 group (1 mL DMEM) were mixed with 1.0 mmol/L H2O2 for 24 h, respectively. While the control group was only treated with 2 mL DMEM solution for 24 h. After the digestion by the trypsin, the 1 mL PBS was added to the cells for the collection of cell supernatants. Th cell supernatant was then incubated with 1 mL DCFH-DA (10 μM) at 37 ℃ for 30 min in the dark. The intracellular green fluorescence intensity was measured at 485 nm excitation and 525 nm emission wavelengths by flow cytometry (Guava® Easy Cyte 6-2L, Milliore, Billerica, USA).

- The authors mention a Figure 4V in the text - it does not exist

Response: Thanks for pointing out our spelling mistakes, we changed the “Figure 4V” to “Figure 4C”.

- in the paragraph: "4.8. Determination of antioxidative enzyme activities in H2O2-induced HepG2 cells", there is practically no information about the conditions of the experiment - what concentrations were tested and how the results were expressed - only short information that kits were used. This point should be better described.

Response: Thanks for your comments. The commercial kits for determining these antioxidative enzymes were purchased from the Nanjing Jiancheng Biotechnology Co., LTD. Thereby, the methods were strictly in accordance with the instructions. We supplemented the detailed information in the section 4.9 as follows (red sections):

Antioxidative enzymes, such as superoxide dismutase (SOD), catalase (CAT), NAD(P)H:quinone oxidoreductase 1 (NQO1) and heme oxygenase-1 (HO-1), play avery important role on scavenging ROS. Thereby, the activities of these antioxidative enzymes were determined in this study. The HepG2 cells (1×105 cells/ mL, 200 μL) were seeded into the 96 well plate for 24 h incubation. The method for cell culture and treatments were the same as described above. The cells were added into 1.0 mL pre-cold PBS and disrupted by an ultrasonic cell disrupter. After centrifugation at 3000 × g for 10 min, the cell supernatant was collected for the investigations of antioxidative enzymes (SOD, CAT, NQO1 and HO-1) according to the instructions of commercial kits (Nanjing Jiancheng Biotechnology Co., LTD).

- ABTS and DPPH methodology descriptions need to be completed; there are no concentrations of the tested samples or prepared reagents, as well as the lack of basic parameters of the assay. It needs to be supplemented.

Response: Thanks for your suggestion, the concentrations for determining the scavenging abilities of ABTS and DPPH were 62.5, 125, 250, 500, 1000 μg/mL, and then the IC50 values were calculated. The specific data is as follows, and relative data was also supplemented in the sections of 4.5.1 and 4.5.2.

ABTS scavenging assay: ABTS free radicals were produced by mixing ABTS stock solution (7 mM) and potassium persulfate (2.45 mM) and the mixture was incubated for 12-16 h in the dark. The ABTS solution was diluted with 0.2 M phosphate buffer (pH 7.4) to an absorbance level of 0.70 ± 0.02 at 734 nm. The ABTS scavenging activity were determined by combining 10 μL 11 DH fractions of DF, EF, BF, WF, and 0%, 20%, 40%, 60%, 80%, 100% fractions (62.5, 125, 250, 500, 1000 μg/mL) with 990 μL ABTS solution. After 10 min reaction, the absorbance was measured at 734 nm.

DPPH scavenging assay: The samples of DF, EF, BF, WF, and 0%, 20%, 40%, 60%, 80%, 100% fractions (62.5, 125, 250, 500, 1000 μg/mL, 1 mL) and DPPH solution dissolved in 95% ethanol (0.1 mM, 1 mL) were mixed by shaking and then incubated at room temperature in the dark for 30 min. The absorbance was determined at 517 nm.

Table 3 The ABTS and DPPH free radical scavenging ability and IC50 values of DH fractions

Free radical

Fractions

Concentrations (μg/mL)

IC50 values

(μg/mL)

1000

500

250

125

62.5

ABTS free radical

DF

94.28±1.26

83.01±1.80

55.85±0.96

39.61±0.84

27.51±0.29

158.87±2.20e

EF

152.15±1.42

98.57±1.69

88.98±1.69

61.43±1.33

26.45±1.95

109.45±2.04h

BF

59.58±2.41

51.77±1.67

35.69±2.34

24.96±1.61

18.10±0.72

534.70±2.73b

WF

30.47±2.61

21.81±1.05

16.19±0.86

12.98±1.07

9.89±1.22

1234.88±3.09a

0%

84.21±2.26

74.93±1.68

50.93±1.03

31.68±1.28

22.13±0.82

220.08±2.34c

20%

84.74±2.28

78.51±1.19

50.45±1.38

34.93±2.28

22.40±0.58

196.22±2.29d

40%

92.08±1.24

89.71±2.93

68.92±1.93

46.70±2.08

33.33±2.26

122.45±2.09g

60%

97.55±1.52

96.47±1.86

79.47±0.54

56.02±2.47

37.79±1.87

94.68±1.98i

80%

99.67±0.52

99.39±1.50

94.88±1.80

62.29±0.67

42.94±2.03

60.49±1.78k

100%

100.17±2.33

99.50±0.53

84.38±1.90

56.17±0.60

35.77±1.35

88.33±1.92j

CF

74.72±0.33

72.76±2.35

70.34±1.11

44.30±1.18

29.13±2.05

152.45±2.18f

Vc

99.77±0.29

85.05±1.38

21.21±1.33l

DPPH free radical

DF

88.22±1.81

70.18±0.60

52.44±1.99

33.65±1.98

25.00±1.17

208.35±2.32g

EF

95.55±0.39

93.98±0.56

82.64±2.70

55.14±1.84

42.48±1.89

87.21±1.94i

BF

54.43±1.39

43.16±1.73

33.26±0.92

26.25±1.26

18.32±0.99

678.18±2.83d

WF

22.11±0.42

16.32±1.39

11.77±1.89

9.33±0.98

7.61±1.20

3681.58±3.57a

0%

48.66±1.13

34.30±0.14

23.02±0.05

16.63±0.14

13.37±0.28

885.66±2.95b

20%

54.29±1.55

45.58±2.92

32.34±0.56

21.09±2.15

14.84±1.01

721.35±2.86c

40%

75.57±2.04

54.75±1.60

40.28±1.23

27.77±1.28

19.65±0.71

350.94±2.55f

60%

87.30±2.15

72.23±1.42

52.59±1.31

35.10±0.49

24.1±1.67

204.95±2.31g

80%

93.17±0.36

89.16±0.72

78.25±0.31

57.63±1.12

37.68±2.01

93.67±1.97h

100%

93.24±0.31

89.36±0.20

79.99±0.81

63.25±1.09

42.44±1.28

78.43±1.89j

CF

69.00±2.36

56.55±1.01

36.90±1.09

23.15±1.05

15.82±1.83

414.33±2.62e

Vc

98.36±1.59

92.03±1.26

17.54±1.24k

Reviewer 3 Report

Dear authors,

Overall, the manuscript is complex and contain new valuable and interesting information’s.

The title of the manuscript accurately describes its subject, and the manuscript includes new information. The abstract is appropriate and the aim of the work clearly established.

There are some negative aspects, which I believe need to be clarified and corrected in order to publish this manuscript. 

Introduction

Line 47 – 49 – “Oxidative stress induced by reactive oxygen species (ROS) is involved in the initiation for many diseases…..”. There are diseases where it contributes to the evolution of the disease, not only initiation.

Results

Figure 1B – is too small to read what appears in the graphics.

Figure 2B – It is not, in my opinion, a suggestive figure for the results. The graphics are illegible and too small to be understood.

Discussions

The discussions are presented in a proper manner, with reference to recent literature data.

Materials and Methods

When it comes to the methodology, I really liked how well-organized everything was and how many tests were performed.

Lines 330 – 344  – For a clearer understanding of fractionation, a schematic presentation (image) could be more helpful.

Line 382 – What extracts or fractions were tested? What levels of DH were present? There are three values given: 50, 100, and 200 mg/mL. What do these concentrations actually signify? Flavonoids or polyphenols? Please elaborate. Generally speaking, this information related to the amounts of additional extracts and particular concentrations must be given with more detail in investigations on cell cultures.

References

The authors used specialized papers that were recently published, and the bibliographic sources they cited provide strong support.

Author Response

Dear Editors and Reviewers:

Thanks for your supports and suggestions, and we have revised our manuscript according to the reviewers’ comments and responded these questions carefully. We hope the reversion may meet the requirements of your esteem journal and reviewers.

Overall, the manuscript is complex and contain new valuable and interesting information’s.

The title of the manuscript accurately describes its subject, and the manuscript includes new information. The abstract is appropriate and the aim of the work clearly established.

There are some negative aspects, which I believe need to be clarified and corrected in order to publish this manuscript. 

Introduction

Line 47 – 49 – “Oxidative stress induced by reactive oxygen species (ROS) is involved in the initiation for many diseases…..”. There are diseases where it contributes to the evolution of the disease, not only initiation.

Response: Thanks for your valuable suggestion, we revised the sentences about oxidative stress in the manuscript (see red section) as follows: Oxidative stress induced by reactive oxygen species (ROS) plays a regulatory role on the initiation, development and evolution of many diseases, such as neurodegener-ative disease, cardiovascular disease, chronic liver disease, type 2 diabetes, aging and so on.

Results

Figure 1B – is too small to read what appears in the graphics.

Figure 2B – It is not, in my opinion, a suggestive figure for the results. The graphics are illegible and too small to be understood.

Response: Thanks for your suggestion, we changed the clearer image of Figure 1B in the manuscript. As for Figure 2B, it is a cell diagram using the flow cytometer. The migration of images to the right means the ROS levels, of which the detection value of ROS was shown as a histogram in the manuscript (Figure 3B). Due to the problem of the worse resolution, we removed this diagram from the manuscript.

Discussions

The discussions are presented in a proper manner, with reference to recent literature data.

Response: Thanks for your suggestion, we reorganize and revised the discussions, and recent references were cited in the manuscript, see red sections.

Materials and Methods

When it comes to the methodology, I really liked how well-organized everything was and how many tests were performed.

Lines 330 – 344  – For a clearer understanding of fractionation, a schematic presentation (image) could be more helpful.

Response: Thanks for your suggestion, we drew a schematic presentation in the 4.2 section and the image is as follows: (see the attachment or manuscript)

Line 382 – What extracts or fractions were tested? What levels of DH were present? There are three values given: 50, 100, and 200 mg/mL. What do these concentrations actually signify? Flavonoids or polyphenols? Please elaborate. Generally speaking, this information related to the amounts of additional extracts and particular concentrations must be given with more detail in investigations on cell cultures.

References: Thanks for your suggestion. After the investigation of various DH fractons on TPC and TFC levels, and scavenging ability on free radicals, 5 fracions (DF, EF, and 60%, 80%, 100% fractions) were selected for the cellular experiments (the relative imformation was introduced in the section 2.4). The IC50 values of scavenging ABTS and DPPH radicals showed that IC50 values’ range of these 5 fractions was from 60 to 208 μg/ mL. Thereby, the concentrations of 50, 100, and 200 μg/ mL were chosed for cellular expreiments. These concentrations represented the concentrations of samples (DF, EF, and 60%, 80%, 100% fractions) (m/v).

The detailed information was supplemented in the method section of cell experiments, see red sections.

The authors used specialized papers that were recently published, and the bibliographic sources they cited provide strong support.

References: Thanks for your recognition.

Round 2

Reviewer 2 Report

Thank you for the thorough manuscript correction and addressing and incorporating the suggested changes. I find that in this form, the article can be published.

Author Response

Reviewer 2:

Thank you for the thorough manuscript correction and addressing and incorporating the suggested changes. I find that in this form, the article can be published.

Response: Thanks for you supports, and your valuable suggestions have guided us to revise our manuscript better. Thank you very much!
